

# **Adaptation tipping points of urban wetlands under a**
# **drying climate**
Amar V.V. Nanda[1,6,*], Leah Beesley[2,6], Luca Locatelli[3], Berry Gersonius[4,6], Matthew
R. Hipsey[5,6], Anas Ghadouani[1,6]
[1] School of Civil, Environmental & Mining Engineering, The University of Western Australia, 35
Stirling Highway, M015, Perth, Western Australia, 6009, Australia amar.nanda@research.uwa.edu.au
[2] Centre of Excellence in Natural Resource Management, The University of Western Australia, PO
Box 5771, Albany, Western Australia, Australia
[3] Department of Environmental Engineering, Technical University of Denmark
Anker Engelunds Vej 1, 2800 Kgs. Lyngby, Denmark
[4] UNESCO-IHE, Westvest 7, 2611 AX Delft, The Netherlands
[5] School of Earth & Environment, The University of Western Australia, 35 Stirling Highway, Perth,
Western Australia, 6009, Australia
[6] Cooperative Research Centre for Water Sensitive Cities (CRCWSC), Clayton, Victoria, Australia
* Corresponding author: School of Civil, Environmental & Mining Engineering, The University of
Western Australia, 35 Stirling Highway, M015, Perth, Western Australia, 6009, Australia
(amar.nanda@research.uwa.edu.au)
Word count: 5,637 (text); 9,022 (text, tables, figures, and references)


**Key Points**
- A modified Adaptation Tipping Point framework is presented to assess the suitability of ecosystem
management when rigorous ecological data are lacking.
- Quantitative boundaries or thresholds to define acceptable ecological change can be overcome by
inclusion of pre-existing thresholds based on available information from policy, legislation, and
involvement of management authorities.
- The extend of legislation, policies, and management authorities across different scales and levels of
governance; need to be understood to adapt ecosystem management strategies.



**Abstract**
Urban wetlands experience considerable alteration to their hydrology, which typically contributes to a
decline in their overall ecological integrity. Wetland management strategies aim to repair wetland
hydrology and attenuate wetland loss associated with climate change. However, decision makers often
lack the data needed to support complex social environmental systems models, making it difficult to
assess the effectiveness of current or past practices. Adaptation Tipping Points (ATPs) is a method that
can be useful in these situations. The method assesses thresholds exceedance of ecological objectives
obtained from policy and informs about the effectiveness of the management strategy to the delivery of
certain social or environmental goals. Here we trial the method on an urban wetland in a region
experiencing a markedly drying climate. ATPs were defined by linking key ecological objectives
identified by policy documents to threshold values for water depth. We then used long-term hydrologic
data (1978-2012) to assess if and when thresholds were breached. We found that from the mid-1990s
declining wetland water depth caused ATPs for the majority of the wetland objectives. We conclude
that the wetland management strategy has been ineffective from the mid-1990s when the region's
climate dried markedly. Empirical verification of the ATP assessment is required to validate the
suitability of the method. However, in general we consider ATPs to be a useful desktop method to
assess the suitability of management when rigorous ecological data are lacking.

**Key words**
Ecosystem management; urban wetland; adaptation tipping points; climate change; management
strategy



## 1. Introduction

Ecological systems with high resilience are able to cope with frequent disturbance and remain
relatively stable over time, whereas systems with low resilience are likely to transition to altered states,
often with reduced function in the wake of disturbance (Holling 1973). Systems with low resilience
can shift between alternative stable states by an incremental change of conditions that induce a
catastrophic (reversible) shift or by perturbations that are large enough to move the system to a lower
alternative state with reduced functions (Scheffer et al. 2001; Folke et al. 2004). Social-ecological
systems (SES) have many functions that depend on feedback mechanisms between processes that take
place at multiple scales (Sivapalan et al. 2012; Elshafei et al. 2014).
Ecosystems are managed to maintain their ecological functions that are particularly vulnerable to
altered processes (e.g. climate change). Such processes can shift ecosystems to reduced ecological
functions (Dudgeon et al. 2006). These complex ecosystems under the influence of drivers of
ecological and social processes can change and then often display nonlinear behaviour with prolonged
periods of stability alternated with sudden changes or critical transitions of the socio-ecological system
(Scheffer et al. 2001; Walker and Meyers 2004). These sudden changes are often not foreseen in
management practices due to its incremental approach which is defined by law enforced threshold
levels along environmental gradients (Walker and Meyers 2004). Interventions to inform policy or
management are therefore not timely or ineffective to maintain ecosystems in a state of prolonged
stability with multiple socio-ecological functions.
Thresholds and tipping points are important focal points for adaptive management (Folke et al. 2005;
Rijke et al. 2012; Haasnoot et al. 2013; Werners et al. 2013), but often lack data to define exact
biophysical thresholds to model complicated interactions in SES models (Schlueter et al. 2012).
However, several indicators (Niemi and McDonald 2004) and 'turning point' approaches do exist that
are commonly used in flood mitigation (Lavery and Donovan 2005; Kwadijk et al. 2010; Reeder and
Ranger 2011; Gersonius et al. 2012), water resources management (Brown et al. 2011; Poff et al.
2015), and institutional adaptation (Lawrence et al. 2013, Fünfgeld 2015) to determine when the
boundaries of a system are reached and future change becomes critical for the system. The method



makes clear what the weak points of the current policy and management are under future stressors to
the system (Hanger et al. 2013).
The turning point approach is also known as 'adaptation tipping point' (ATP) method. ATPs are
reached if the magnitude of change is such, that a current management strategy can no longer meet its
objectives (Kwadijk et al. 2010). As a result, adaptive management is needed to prevent or postpone
these ATPs. This method was recently applied to a species re-introduction program and assessed how
long the socio-ecological baseline strategy remained effective under future climate change (Bölscher et
al. 2013; Werners et al. 2013; van Slobbe et al. 2016). The timing of an ATP does not necessarily
correspond to ecological or social tipping points (Werners et al. 2013). However, the ATP approach
confronts the lack of quantitative and qualitative ecological data sets to infer acceptability of
management (Wardekker et al. 2010; Haasnoot et al. 2012; Haasnoot et al. 2013) by stakeholder
engagement to determine unknown/ill-defined thresholds and prevents a focus on only existing
management strategies (Wardekker et al. 2010; Bölscher et al. 2013). To prevent confusion with
definitions of tipping points in the other fields (e.g. climate sciences, ecology) we will use 'adaptation
tipping point' in our study.
A deficiency of the ATP methodology is the understanding how an ecosystem management strategy
compares to ecological resilience when detailed models or sufficient data are unavailable.  In other
words: the management strategy needs to be informed about when the ecosystem could shift into an
alternative state with low resilience when the system is exposed to stressors which are induced by
climate change. Wetlands are ecosystems that are particularly vulnerable to decreased ecological
resilience due to altered hydrology, invasive species, nutrient loading, fire regimes etc. that can cause
wetlands to shift from a 'clear-water' stable state to a 'turbid-water' stable state or from a permanently
to a seasonal hydro-regime that inadequately supports ecological processes (Scheffer et al. 2001; Folke
et al. 2004).
The wetland in our case study area is located in the biodiverse region in south-west Western Australia
(Myers et al. 2000) and has been noticeably impacted by anthropogenic factors (Bekle 1981; Bekle and
Gentilli 1993). An estimated 85% of the Swan Coastal Plain (SCP) wetlands have been lost since
colonial settlement and are likely to experience increasing hydrological stress due to further decreasing



rainfall (Balla 1993; Davis and Froend 1999). The altered hydrology of wetlands in Perth is likely to
have breached natural ecological tipping points and caused environmental degradation unless the
wetlands are highly resilient. The key challenge to the catchment's wetland management is to adapt to
this drier regime while climate change predictions and the ecological response is uncertain. Therefore,
the catchment area is suitable to apply the ATP methodology to determine whether the current wetland
management strategy is effective to prevent undesirable ecological tipping points.
We are interested as to when and how much hydrological variation an ecosystem can cope with before
the durability of a strategy to conserve the ecosystem expires. The overall aim of this study is to
provide a modified ATP framework to identify the effectiveness of ecosystem management strategies.
We define effectiveness by three aspects of the ecosystem and subdivide this into three aims to
identify:
1) the hydrological response and variation of the ecosystem under climate change by conducting a

121       literature study and by interviews with experts;

2) temporal scale and ecosystem responses with the determination of ATPs in hydrologic time-series

123       data for each socio-ecological objective from the wetland management strategy;

3) the recovery rate or alternative stable state of ecological processes that is defined by minimum and

125       maximum water-level thresholds compared to objectives mandated by policy and management.





**2. Method**

**2.1. ATP method and case study area**
The original five-step ATP methodology include (Figure 1): (i) the determination of climate change
effects on the system; (ii) followed by identifying key objectives and thresholds; (iii) the determination
when standards were compromised in the past; (iv) analysing when standards were compromised in the
future; and (v) to repeat step 1-4 for alternative strategies. Further details about the original
methodology can be found in Kwadijk et al. (2010). We modified the original methodology to a three-
step assessment as we only assess historical time series. Firstly, we assessed the observed hydrological
changes followed by determining objectives and thresholds. At last, we combined step 3 and 4A of the
methodology to interpreted ATPs in conjunction with understanding the ecosystem processes,
feedbacks, and alternative stable states (Figure 1).
This study assessed one wetland, Forrestdale Lake (Figure 2), which is located in the biodiverse region
of the Swan Coastal Plain in south-west Western Australia (Myers et al. 2000). The wetland supports
many waterbirds and its surrounding riparian vegetation supports terrestrial birds, significant reptiles,
mammals, and other vertebrate species (Balla 1993). The lakes' high biodiversity makes it an
important regional conservation area (CCWA 2005). Since colonial settlement, the lake has been
exposed to several stressors, such as land-use changes, urban encroachment, nutrient run-off, and
decreasing surface water levels in the lake.
Similarly to other Mediterranean regions in the world, the south-west of Western Australia is
experiencing reductions in rainfall that lead to decreasing recharge of the aquifer (Petrone et al. 2010).
Approximately 80% of the annual precipitation occurs in winter between May and September, with
groundwater recharge occurring from June to September (DoW 2008). The wetland experiences a
Mediterranean climate with a mean annual rainfall of 852 mm in the period 1980-2014. Since the
1970s this region has experienced a 10-20 % decrease in average annual rainfall that resulted in a mean
annual rainfall of 775 mm in the period 2004-2014 (Charles et al. 2010; Smith and Power 2014).
Despite high resilience, the wetland shows a rapid decline of its critical ecological processes as a result



of less surface water availability (Froend et al. 1993; Balla and Davis 1995; Sommer and Horwitz
2009; Sommer and Froend 2011). These observed impacts of climate change on the hydrology make
the wetland a suitable study area to apply the ATP method.

**2.2. Data collection and analyses**

**2.2.1. Step 1: Legislative framework and impacts of climate change - literature review**
The scope of the assessment is defined in line with the legislative basis for Forrestdale Lake. In
Western Australia, the Environmental Protection Act (1986) is the legislative act that underpins the
environmental protection of wetlands. According to the EP Act, the 'Ministerial water requirements for
the Gnangara Mound and Jandakot wetlands' (1992) mandates ecological water requirements that
consist of upper and lower thresholds to maintain ecological processes. Protection of biodiversity or
conservation values such as maintaining biodiversity is included in the Conservation and Land
Management Act (1984) and the Wildlife Conservation Act (1950). Large regional wetlands have also
been listed as Ramsar (e.g. Forrestdale Lake) to protect waterbirds (Ramsar 1994) and to protect
migratory birds under several international agreements (JAMBA 1981; CAMBA 1988; ROKAMBA
2006). Protection of nationally and internationally important flora, fauna, and ecological communities
is arranged by the Commonwealth of Australia under the Environment Protection and Biodiversity
Conservation Act (EPBC 1999). The above mentioned Acts and Agreements provide the statutory base
to formulate wetland management plans. A preceding wetland management plan from 1993 for
Forrestdale Lake was updated in 2005 which includes the ecological values of the wetland; proposes
management actions to control invasive species; and mentions the risks of declining water levels
(CCWA 2005).
Climate change, via its impact on rainfall and groundwater recharge, is an important regional driver of
wetland hydrology and ecological functions (Eamus and Froend 2006; Barron et al. 2013). Local-scale
hydrologic changes associated with land-use change and groundwater abstraction may also impact
water levels of wetlands. Although, these changes are considered minimal compared to region-wide



changes in rainfall and consequently recharge of the aquifer (Townley et al. 1993; McFarlane et al.
2012). There is evidence that climate change is impacting the hydrology of the unconfined aquifer
since the 1970s (Froend et al. 1993; Davis and Froend 1999; Froend and Sommer 2010; Ali et al.
2012) which is likely to continue during the 21st century (Charles et al. 2010; Smith and Power 2014).
In Figure 3 we represent the rainfall decline and population growth of Perth which resulted in growing
water demand while groundwater availability is declining (ABS 2014).
Changes in the hydrology were noticeable from the end of the 1980s after the rainfall reduction in the
1970s. Prior to the 1950's the wetland was classified as a 'groundwater through flow lake', but is now
considered as a 'permanently inundated and perched lake' depending on rainfall and groundwater
(Semeniuk 1987; Hill 1996; Dawes et al. 2009). However, recently a combination of disconnection
from groundwater and decreasing annual rainfall resulted in a lake that is seasonally inundated
(CCWA 2005). In Figure 4 we present a timeline of the legislation framework with key social and
environmental events that have occurred in Forrestdale Lake and its groundwater catchment area.

**2.2.2. Step 2: Select objectives and quantify threshold values - literature review**
In the second step, we reviewed the current wetland management strategy for policy objectives,
indicators, and threshold values of the wetland ecological processes. These functions represent the
critical objectives of the wetland management strategy. Certain water depths are needed within a
wetland to sustain a variety of ecological processes (Froend et al. 2004; Eamus and Froend 2006;
Canham 2011; Barron et al. 2013). Therefore we used water depth as a proxy to link ecological
objectives to acceptable thresholds. We identified two pathways within the SES via which water depth
can impact on wetland ecological objectives:

1. Water depth may reach levels that are too low:
(a) to maintain sediment processes
(b) to provide habitat needed by waterbirds, frogs, freshwater turtles, and macro-invertebrates for
survival and reproduction
(c) that lead to increasing weed invasion to compromising habitat needed for wading birds




(d) that inhibit the growth of mosquitoes and midges
2. Water depth may reach levels that are too low or too high, such that they lead to:
(a)  the death of phreatophytic and fringing vegetation.
(b)  compromising habitat needed for terrestrial birds and mammals

From the aforementioned pathways, we derived eight critical ecological objectives, see Table 1. The
objectives were taken from the Forrestdale Lake wetland management strategy (CCWA 2005); the
Ministerial water requirements (EPA 1992), and were discussed with two experts from different
management authorities (Department of Parks and Wildlife and Department of Water). For each
ecological objective, minimum water depth requirements were obtained (i.e. threshold) using the
Ministerial water requirements (Table 1). In cases where water level thresholds were not informed by
the Ministerial water requirements, we relied on peer-reviewed literature (See 'Source' in Table 1). A
detailed description of necessary conditions can be obtained from previous research (Balla 1993;
Storey et al. 1993; Balla and Davis 1995; Froend et al. 2004; Dale and Knight 2008; Department of
Environment and Conservation 2011). In addition, two expert interviews were conducted to determine
the accepted exceedance frequency and to define threshold definitions that were not informed by
policy or literature. The appraisal of the ecological objectives in Table 1 reveals that 21.6 mAHD is the
minimum threshold for vegetation (Townley et al. 1993), mammals, and terrestrial birds; and 22.0
mAHD is the minimum threshold to maintain waterbirds, freshwater turtles, frogs, and macro-
invertebrates.

**2.2.3. Step 3: Determine ATPs - statistical analyses**
We observed time series of surface and groundwater depths (Site ID 14578 and 12781400) as provided
by the Department of Water's water information database (DoW 2015). As the lake experienced
hydrological change during the 1990s, the data set was divided into two time periods 1978-1995 and
1996-2012. To evaluate the ecological resilience of the wetland, we assessed when and for how long
the water level in Forrestdale Lake crossed the thresholds. For the calculation of threshold exceedance
we used the observed (historical) time series of water levels to estimate the frequencies of occurrence



of threshold exceedance by annual minimum series (Jenkinson 1955). Equation 1 describes the
distribution $G(x)$ of the magnitude of events $x$ smaller than a threshold $x_0$ over a (non)-consecutive
period of time over a period of years $T$. Here $\alpha$ and $k$ are constants derived from the average highest
and lowest in sets of $T$ annual minima and the minimum value to be expected once in $T$ years. To
interpret the occurrence of ATPs in context with the ecological tipping points; we extended our
analyses by comparing the drought frequency, duration and start month for both the pre and post 1995
water-level time series. A dry period was considered when water depth was lower than 21.6m. for 3
consecutive months. We compared the water levels with the available historical ecological data to
make an estimation of the trajectories over time.

$G(x) = 1 - \left[ 1 - k \left( \frac{x - x_0}{\alpha} \right) \right]^{\frac{1}{k}}$ for k ≠ 0  (eq. 1)
**3.0. Results**

The results are represented in three steps in accordance with our methodology, as per Figure 1. The
first two steps show the results of the literature review and step 3 shows the results from time series
analyses of historical surface and groundwater level data from 1978-2012. From the literature review,
we revealed that protection of the regional important Forrestdale Lake wetland lake is provided by
legislation and policies on different levels and scales (Figure 5). The management of the lake is
therefore organised on different levels of government institutions that have their own scale of
operation (e.g. local council vs. state wide department). Due to the different institutions and their
operational level, the execution of the wetland management strategy is a shared responsibility of all
stakeholders. However, the co-ordination of this strategy is the responsibility of a state-wide operating
institution (Department of Parks and Wildlife). System controls (e.g. policy and legislation) are
mandated on a larger spatial scale, whereas accumulated stressors (e.g. reduced rainfall or lowering
groundwater table) have larger impacts on a lower spatial scale, such as on ecosystem scale or separate
ecological processes of the ecosystem. These noticeable effects are translated by threshold exceedance
of ecological processes.





From the extensive variety of policies and legislation in place to protect the ecological values of the
wetland we were able to derive the important socio-ecological objectives for the wetland. For each
objective, we determined the critical water requirement thresholds. Although for our analyses the water
requirement policies did not provide maximum exceedance frequencies (return period) for each
objective. Where return periods for certain objectives in the management strategy were lacking,
stakeholders were able to provide expert knowledge to determine threshold definitions, such as for
drought duration; water availability for birds, and exposure of acid sulphate soils.
We found from the expert interviews that legislation and policy aims are a good starting point to
discuss with stakeholders that operate on a state wide scale. These experts represent management
authorities that are responsible for execution of larger scale (top-down) policies and legislation. Data
of monthly observed surface and groundwater levels in the lake were available and publicly accessible
via the State's Data Portal. Groundwater level data is only available from 1997 and surface water
levels from 1952. Surface water level from the start of the observations until 1978 contains many data
gaps to adequately perform ATP analyses.
A combination of a review of peer-reviewed literature and government reports provided a complete as
possible overview of ecological studies undertaken in Forrestdale Lake. Data are predominantly
available in government reports rather than in peer-reviewed media. This included data on bird counts,
macro-invertebrates species composition, and vegetation transects. Ecological data is often patchy and
only available for certain time frames in the 1990s and 2000s for Forrestdale Lake when requested
from government departments. Bird counts for the lake have been discontinued since 2009 (DoW
2012) and vegetation transects are not conducted on regular basis as mandated in policy.
ATPs were determined by calculating the re-occurring water level depth using the values from Table 1
with Equation 1. The ATP analysis employed here suggests that a drying climate has compromised
four ecological objectives of Forrestdale Lake (Table 2). ATPs occurred after 1995 and threshold
crossings occurred for vegetation and mammals, waterbirds, turtles, macro-invertebrates. Water levels
for remaining objectives are close to exceeding thresholds such as the capacity of the lake to deliver
sediment processes and limiting the risk of oxidation of acid sulphate soils in the lake bed.


When the drought frequency and duration are compared for both periods, before and after 1995, we see
major differences (Figure 6). Prior to 1995, no dry periods of 3 consecutive months occurred, however,
the lake did dry completely five times for at least one month. These five occurrences are not
considered as a drought according to our definition of 3 consecutive months. In figure 5 we have
included the dry periods prior to 1995 to compare the duration and start month of each drought. The
drought frequency is 5x before 1995 (definition 1 month/year) and increases to 16x after 1995
(definition 3 consecutive months). From Figure 6 we observe that Forrestdale Lake dried more
frequent than the recommended return period of 1 in 5 years and that each dry period exceeded the
maximum duration of 3 consecutive months. Drying is most frequent in summer months December,
January and February which is in contrary to regulation that drying of the lake should not occur before
May in order to ensure waterlogged lake bed throughout the year and limited water availability for
species.
Although there is not enough data to conduct trend analyses we observe a large increase in the
frequency of droughts and duration of each drought after 1995 compared to prior 1995. When we
combine the results from our ATP analyses (Table 2) with the drought analyses (Figure 6), we observe
a regime shift in the ecosystem from a permanently to a seasonally inundated wetland. The effect of
this hydrological shift translates into passing the defined threshold level that is enforced in policy and
leading to an ATP. In Figure 7 we graphically present the minimum thresholds for all objectives; the
water levels from 1978-2012 compared to the initiation of groundwater abstraction; and the
implementation of the water policy requirements.
Compared to the implementation date of the water requirements policy in 1992; water level
exceedance for ecological objectives occur in the period after the water policy was implemented.
Between the 1970s and the implementation period of the policy in 1980s no significant research was
conducted on the gradual decline of water levels in the Swan Coastal Plain wetlands. With available
quantitative ecological data on ecological responses we base our representation with stylised lines to
explain individual ecological responses compared to declining water levels from the 1970s. This
representation is a combination of historical data from previous research and information from expert
interviews. The decline of the ecological processes is simultaneous with the increased duration and



frequency of dry periods during the 1990s. While minimum water requirements for the wetland were
not updated in the state water requirements policy since its introduction in 1992; existing water
requirements were used in 2005 to determine the current wetland management strategy. After the mid-
1990s we observe that the management does not respond to maintain declining water levels on the
mandated threshold levels.
**4.0. Discussion**
**4.1. Temporal and spatial hydrological responses in ATP analysis applied to ecosystems**
A major gap in the science-policy interface and socio-hydrologic systems literature is: (i) the
identification of inadequate policy to inform managers or policy makers about the durability of an
ecosystem management strategy; (ii) to perform assessments of hydrological variables when data is
lacking. With the ATP methodology presented we have tried to further close this gap in the literature.
The methodology presented, assessed whether a baseline ecosystem management strategy was
sufficient to sustain the ecological resilience of the ecosystem. Our ATP framework assesses resilience
of the hydrological system across spatial and temporal scales by (Zevenbergen et al. 2008): (i) the
amount of reaction of the ecosystem; (ii) the temporal scale and ecosystem responses to increased
perturbations; and (iii) the recovery rate or by a shift from a desirable stable state to an alternative and
undesirable stable state with limited ecological processes.
The observed climatic shift by the end of the 1960s and early 1970s in south-west Western Australia
(Verdon-Kidd et al. 2014) follows the stepwise decreasing rainfall trend in our hydrological time
series. We observe a hydrological response in the 1990s with shorter periods of inundation and ATPs
occurring simultaneously in the same time period. Other studies explain this hydrological shift from
permanent to intermittent water availability in the lake by decreased surface water availability due to
lower rainfall (Eamus and Froend 2006; Davis and Brock 2008; Dawes et al. 2009; Maher and Davis
2009). Our observations of consistent reductions of water levels result in more frequent, prolonged dry
periods. Studies confirmed that a significant reduction in water levels for consecutive years could
threaten the regional function of wetlands to sustain multiple ecological functions (Froend et al. 2004;
Davis and Brock 2008; Maher and Davis 2009).



The analysis points to an ineffective water requirements policy as water levels are exceeded for four of
the eight ecological functions. Thresholds were crossed in the 1990s which occurred simultaneously
with the observed hydrological response. The main ecological processes depend on waterlogged soils
during low water availability but are at increasing risk when the lake bed dries completely over
summer. Late drying of the lake does imply a lack of surface water availability for species that have a
limited action radius to alternative habitats, such as macrophytes, freshwater tortoises, frogs, and
macro-invertebrates. Our study did not include the investigation of ecological responses. However, the
hydrological change and ATPs are followed by declining trends in the ecology that was showed in
more recent studies through:
- increasing weed invasion and exotics establishing in the understory and deterioration of fringing

357       vegetation (Froend et al. 2004; Davis and Brock 2008)

- a gradual declining trend in the species numbers and composition of macro-invertebrates (Balla and

359       Davis 1995; Maher and Davis 2009; Sommer and Horwitz 2009).

- decreasing numbers of birds from over 20.000 birds in the 1980s (Storey et al. 1993; Maher and

361       Davis 2009) to just over 10.000 birds in 2009 (Bamford et al. 2010).


Literature describes the responses of ecosystems after perturbations and the shifts that could occur
likely to shift from a desirable higher stable alternative state into a undesirable lower alternative stable
state with high resilience and reduced ecological processes (Scheffer et al. 2001; Folke et al. 2004;
Folke et al. 2005). Due to a lack of data to determine shifts between multiple or alternative stable states
(Capon et al. 2015); our analyses combines rapid hydrological processes and slow response of
ecological processes such as vegetation (Sivapalan and Blöschl 2015) under the influence of an
external boundary condition (lower rainfall due to climate change). Different to regime shifts in the
natural system that trigger a shift in the social system to restore environmental degradation (Elshafei et
al. 2016); a gradual transition appears not to trigger management interventions to maintain the rapid
processes in an ecosystem. The understanding of scale and level of policy and legislation that provide
the legislative framework of management practises is critical, since this could enhance or constrain the
necessary shift in the social system.





### 4.2. Informing ecosystem management

The presented framework provides in an early stage guiding principles to existing ecosystem management strategies when these are ineffective. The ineffectiveness of current policy and management were also shown in flood risk studies (Lavery and Donovan 2005; Reeder and Ranger 2011), flood mitigation under climate change (Gersonius et al. 2012), for river restoration (Bölscher et al. 2013) and for the impact of the hydrological regime of a river on salmon re-introduction and shipping (van Slobbe et al. 2016). Central in these studies is to determine *when* and *how much* action is needed to determine alternative management strategies (Sivapalan and Blöschl 2015) In the interest of decision makers or managers ATPs are used as a starting point to explore adaptation measures that adequately resolve the critical adaptation tipping point (Hanger et al. 2013) when quantitative data is not readily available to support a complex model with predicted feedback mechanism in the socio-environmental system (Sivapalan et al. 2012; Di Baldassarre et al. 2013; Elshafei et al. 2014; Di Baldassarre et al. 2015). However, rather than substituting existing quantitative assessments in the socio-hydrology; the outcomes of an ATP analyses provide better understanding of the role of individual processes before making more complex models (Hipsey et al. 2015) and highlights the potential dynamics of scale of legislation, policy and interaction of management authorities in the hydrological system.

To adequately inform existing management practices, we first consider the whole set of clearly stated objectives in a management strategy without prioritising or aggregating these. As a result, we provide the alternative states of ecological processes within the spatial and temporal scales of processes and governance systems (Niemi and McDonald 2004). Studies showed that introducing multiple management aims overcomes a focus on separate ecological objectives that lead to a lack of quantitative boundaries or thresholds for acceptable ecological change (Hallegatte 2009; Kwadijk et al. 2010; Haasnoot et al. 2012; Werners et al. 2013). Studies showed that when defined threshold levels along an environmental gradient are passed which are enforced by law (Walker and Meyers 2004); not all ecological processes would show a direct decline of species or shift in species composition. From a management perspective reversing the ecosystem to a stable state with adequate ecological processes involves measures that need to be far enough to reverse the conditions the ecosystem (Scheffer et al.





2001). Therefore, informing decision-makers at an early stage prevents costly measures to reverse the
system.
Secondly, in the absence of clearly defined thresholds our framework provides active involvement of
the management authorities (Haasnoot et al. 2012; Haasnoot et al. 2013) from a multi-purpose
perspective (van Slobbe et al. 2016). The ATP analyses stimulate stakeholders to look at the durability
of their approach (Kwadijk et al. 2010). Continuous improvement in the processes of adaptive
management is an ongoing challenge. Studies demonstrate frameworks for collaborative research in
the science-policy interface across several scales (Mitchell and Hollick 1993; Davis et al. 2015).
Threshold definitions for management approaches also reflect the ideas of multiple management
authorities when management practices needs to be updated. Without a coupled system there is still
potential to provide insight into the impact of management interventions by capturing the combined
measures to adapt the current strategy.

### 416     4.3. Adapting management strategies

For effective governance developing a better understanding of climate and hydrological impacts is
required (Davis et al. 2015). With the involvement of stakeholders in our assessment we can account
for the exploration of future hydrological events and provide decision-makers time periods for when
the expiry of current policies occur. The ATP assessment includes the option to identify measures and
for adequate governance decisions, further exploration of adaptation measures under future climate
scenarios needs to be investigated. This could include: 1) physical/engineered measures; 2) adoption of
new or amended policy instruments; 3) adoption of policy strategies (combination of options 1 and 2);
or 4) implementation of an adaptation strategy (Folke et al. 2005; Nelson et al. 2007; Kwadijk et al.
2010). Critical to successful adaptation requires understanding the scale and level of implementation of
existing policies, legislation or management strategies that are often barriers to local scale adaptation.
Our ATP analysis shows that the ecosystem management strategy is not designed to cope with current
hydrological variation. The application of the proposed methodology is adequate for ecosystems:
without clear boundary conditions and defined thresholds; external drivers that cause regime changes
over time; and a rapid assessment is required to provide overview of *when* management strategies are



ineffective and to *which* failing objectives interventions can be taken. However, the limitations of the
study include the effects of multiple stressors on the system; a limited focus on new strategies; and
including objectives or thresholds that change over time due to socio-economic changes.
**5. Conclusion**

The extended ATP method presented in this paper provides a combination of a qualitative and
quantitative analysis of datasets of a wetland ecosystem. We applied the concept of 'adaptation tipping
points', to identify when management response became inadequate to prevent decline in ecological
integrity. Through a combination with conceptual and visual representation of the ecological processes
we proved to be able to identify major trends and transitions in the system in the presence of strong
driver of change and variable hydrological conditions.
This approach was useful to determine the effectiveness of an ecosystem management strategy when
data availability is limited and social-ecological dynamic models to fully assess the tipping point and
potential points for interventions are absent to monitor suitability of management. This study showed
that a lack of data, quantitative boundaries or thresholds to define acceptable ecological change can be
overcome by inclusion of pre-existing thresholds based on available information about shifts of the
wetland's hydrological regime. This included information about unacceptable adverse ecological
changes to the unique set of identifiers, and the input of expert knowledge to determine the critical
wetland objectives and thresholds for wetland management. We showed in an early stage information
to stakeholders to determine the effectiveness of existing wetland policy that can be used to adapt or
accept objectives, thresholds; seen in context with ATPs and undesirable ecological changes. With the
absence of SES models the ATPs of underlying ecological processes were seen in relation to
undesirable ecological responses. ATPs could establish a proxy indicator for lag-responses in the
ecology to timely adapt ecosystem management before ecological processes exceed unaccepted levels.

**Acknowledgements**



The authors thank the Department of Parks and Wildlife and the Department of Water that provided
the ecological and water level data of Forrestdale Lake. The RStatistics code to compute the water
level data was provided by Ms. Chrianna Bharat (The University of Western Australia). This research
was conducted within program B4.2 of the Cooperative Research Centre of Water Sensitive Cities.
Amar Nanda would like to acknowledge the PhD scholarship funding provided by the Scholarships for
International Research Fees (SIRF) funded by The University of Western Australia.





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

Contributions of a Resilience Framework." Annual Review of Environment and Resources 32(1): 395-

631    419.

Niemi, G. J. and M. E. McDonald (2004). "Application of Ecological Indicators*." Annual Review of
Ecology, Evolution, and Systematics 35(1): 89-111.
Petrone, K. C., J. D. Hughes, T. G. Van Niel and R. P. Silberstein (2010). "Streamflow decline in
southwestern Australia, 1950-2008." Geophysical Research Letters 37(11): n/a-n/a.
Poff, N. L., C. M. Brown, T. E. Grantham, J. H. Matthews, M. A. Palmer, C. M. Spence, R. L. Wilby,
M. Haasnoot, G. F. Mendoza, K. C. Dominique and A. Baeza (2015). "Sustainable water management
under future uncertainty with eco-engineering decision scaling." Nature Clim. Change advance online
publication.
Ramsar (1994). The Ramsar Convention on Wetlands , as amended in 1982 and 1987, Ramsar
Convention Secretariat, Gland, Switzerland: 5-5.
Reeder, T. and N. Ranger (2011). "How do you adapt in an uncertain world?: lessons from the Thames
Estuary 2100 project."
Rijke, J., R. Brown, C. Zevenbergen, R. Ashley, M. Farrelly, P. Morison and S. van Herk (2012). "Fit-
for-purpose governance: A framework to make adaptive governance operational." Environmental
Science & Policy 22: 73-84.
ROKAMBA (2006). "Agreement between The Government of Australia and The Government of The
Republic of Korea on The Protection of Migratory Birds." Australian Treaty Series 2007(ATS 24).
Scheffer, M., S. Carpenter, J. A. Foley, C. Folke and B. Walker (2001). "Catastrophic shifts in
ecosystems." Nature 413(6856): 591-596.
Schlueter, M., R. McAllister, R. Arlinghaus, N. Bunnefeld, K. Eisenack, F. Hoelker, E. MILNER-
GULLAND, B. Müller, E. Nicholson and M. Quaas (2012). "New horizons for managing the
environment: a review of coupled social- ecological systems modeling." Natural Resource Modeling

654    25(1): 219-272.





Semeniuk, C. A. (1987). "Wetlands of the Darling System- a geomorphic approach to habitat
classification." Journal of the Royal Society of West Australia 69: 95-112.
Sivapalan, M. and G. Blöschl (2015). "Time scale interactions and the coevolution of humans and
water." Water Resources Research 51(9): 6988-7022.
Sivapalan, M., H. H. G. Savenije and G. Blöschl (2012). "Socio-hydrology: A new science of people
and water." Hydrological Processes 26(8): 1270-1276.
Smith, I. and S. Power (2014). "Past and future changes to inflows into Perth (Western Australia)
dams." Journal of Hydrology: Regional Studies 2: 84-96.
Sommer, B. and R. H. Froend (2011). "Resilience of Phreatophytic Vegetation to Groundwater
Drawdown: Is Recovery Possible Under a Drying Climate?" Ecohydrology 4(1): 67-82.
Sommer, B. and P. Horwitz (2009). "Macroinvertebrate cycles of decline and recovery in Swan
Coastal Plain (Western Australia) wetlands affected by drought-induced acidification." Hydrobiologia

667 624: 191-203.

Storey, A. W., R. M. Vervest, G. B. Pearson and S. A. Halse (1993). Wetlands of the Swan Coastal
Plain, Volume 7, Waterbird Usage of Wetlands on the Swan Coastal Plain. Perth, Western Australia,
Water Authority of WA.
Townley, L., J. Turner, A. D. Barr and M. Trefry (1993). Wetlands of the Swan Coastal Plain Volume
3: Interaction Between Lakes, Wetlands and Unconfined Aquifers, Education Department of Western
Australia.
van Slobbe, E., S. E. Werners, M. Riquelme-Solar, T. Bölscher and M. T. H. van Vliet (2016). "The
future of the Rhine: stranded ships and no more salmon?" Regional Environmental Change 16(1): 31-

676 41.

Verdon-Kidd, D. C., A. S. Kiem and R. Moran (2014). "Links between the Big Dry in Australia and
hemispheric multi-decadal climate variability – implications for water resource management."
Hydrology and Earth System Sciences 18(6): 2235-2256.
Walker, B. and J. A. Meyers (2004). "Thresholds in ecological and social–ecological systems: a
developing database." Ecology and Society 9(2): 3.





Wardekker, J. A., A. de Jong, J. M. Knoop and J. P. van der Sluijs (2010). "Operationalising a
resilience approach to adapting an urban delta to uncertain climate changes." Technological
Forecasting and Social Change 77(6): 987-998.
Werners, S., S. Pfenninger, E. van Slobbe, M. Haasnoot, J. Kwakkel and R. Swart (2013). "Thresholds,
tipping and turning points for sustainability under climate change." Current opinion in environmental
sustainability 5(3): 334-340.
Werners, S., R. Swart, E. van Slobbe and T. Bölscher (2013). "Turning points in climate change
adaptation." Global Environmental Change 16(3): 253-267.
Wildlife Conservation Act (1950). "Wildlife Conservation Act 1950." Government of Western
Australia 077 of 1950: (14 & 15 Geo. VI No. 77).
Zevenbergen, C., W. Veerbeek, B. Gersonius and S. Van Herk (2008). "Challenges in urban flood
management: travelling across spatial and temporal scales." Journal of Flood Risk Management 1(2):

694    81-88.






Figure 1 The complete Adaptation Tipping Point methodology with an overview of the steps
undertaken in this study (blue boxes) and the according data collection and analyses (Adapted from:
Kwadijk et al. 2010).
Figure 2 Location of Forrestdale Lake (32° 09' 30" S; 115° 56' 16" E) within its groundwater
catchment with the indication of increasing urbanisation in the catchment; the multiple management
authorities; and protection policies (map projection GDA94).
Figure 3 Growing water demand caused by population growth (ABS 2014) in Perth with decreasing
water availability as a result of decreasing rainfall (BoM 2016).
Figure 4 A historical representation of time and scale the traditional human-nature system and water
resources system of Forrestdale Lake with indicated key events of the four subsystems: natural
resources, infrastructure, socio-economics and institution.
Figure 5 Ecological resilience and legislation: across spatial levels, shows large scale impacts through
the catchment that accumulate and result in exceedance of thresholds for ecosystem services.
Figure 6 Comparison of the onset and duration of drought from 1978-2012 at Forrestdale Lake prior
and post 1995. Each bar represents a dry period which is defined as 1 month per year (post 1995) and ≤
3 consecutive months (post 1995).
Figure 7 Ecosystem regime shift on the onset of dry periods with declining water levels and the change
of conditions of ecological processes over time. Incremental management and policy compared to non-
linear ecosystem responses over time are ineffective when sudden changes occur.





Table 1 Threshold values for the ecological objectives to determine ATPs for surface water (SW) and
groundwater (GW) levels in (non)-consecutive months, represented as the mean water level in
Australian Height Datum in meters (mAHD).
Table 2 Adaptation tipping points (1 in 5 years exceedance water depth (m) calculated with eq. 1) for
each ecological function of Forrestdale Lake with red indicating an ATP has occurred and green not
occurred.






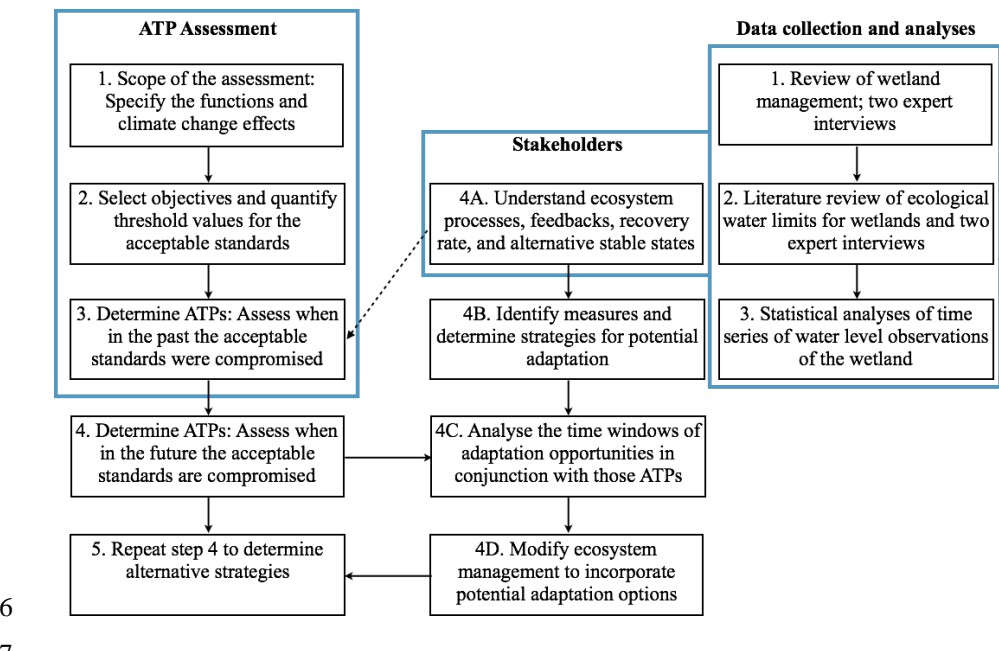








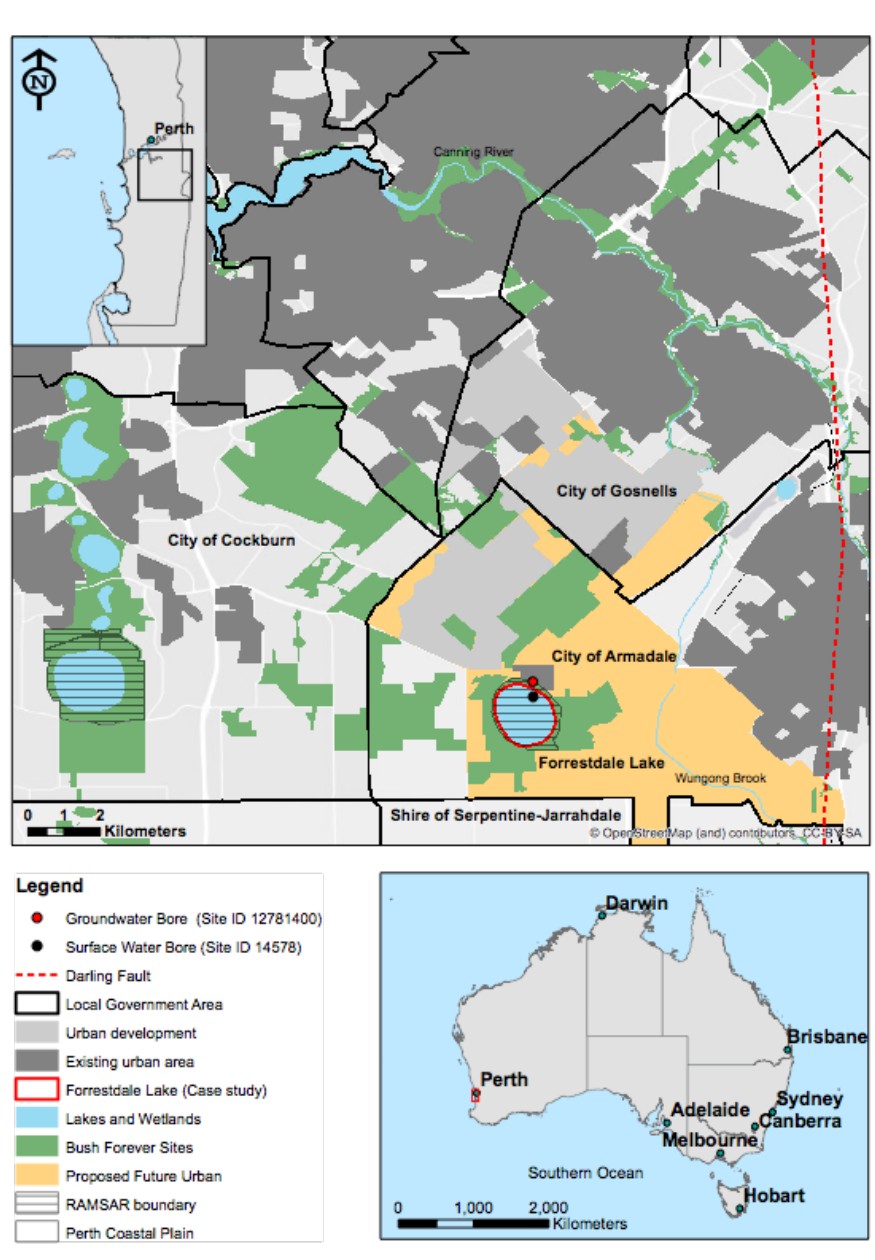









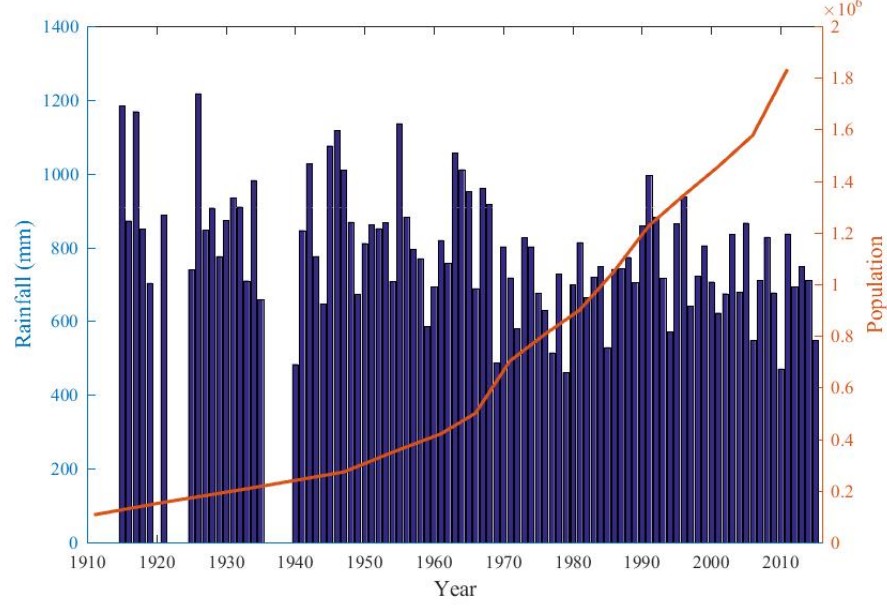






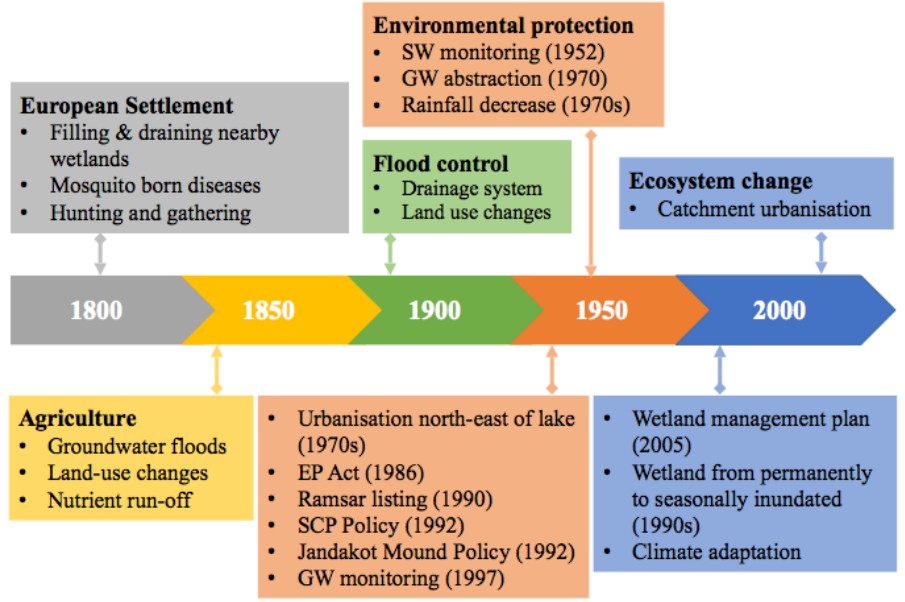










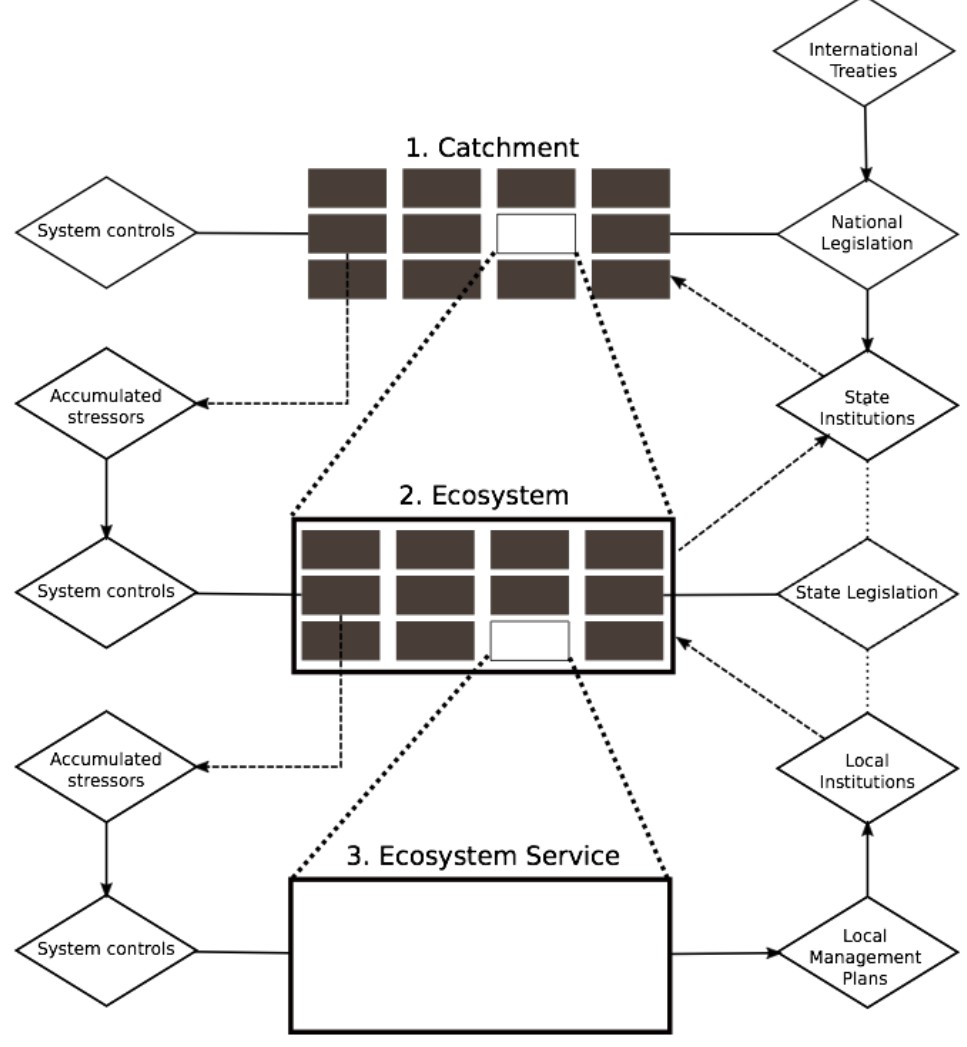






| Ecological objectives | Water level (mAHD) | Threshold definition | Source |
|---|---|---|---|
| 1. protect vegetation and mammals; definition of drought | SW < 21.6 | 3 consecutive months; 1 in 5 years | EPA (1992); Froend et al. (2004); CCWA (2005) |
| 2. prevent mosquitoes | SW < 21.6 | 1 month per year; 1 in 1 year | CCWA (2005) |
| 3. protect waterbirds | SW < 21.6 | 6 consecutive months; 1 in 5 years | EPA (1992); Storey et al. (1993); CCWA (2005) |
| 4. protect frogs | SW < 21.6 | 8 months; 1 in 5 years | Froend et al. (2004); CCWA (2005) |
| 5. protect tortoises | SW < 21.6 | 3 months; 1 in 5 years | Froend et al. (2004); CCWA (2005) |
| 6. protect macro-invertebrates | SW < 22.0 | 3 consecutive months; 1 in 5 years | Froend et al. (2004); CCWA (2005) |
| 7. prevent exposure of Acid Sulphate Soils | GW < 21.1 | 3 consecutive months; 1 in 5 years | Froend et al. (2004) |
| 8. maintain sediment processes | GW < 21.1 | 3 consecutive months; 1 in 5 years | Froend et al. (2004) |





| Ecological objective | Threshold | Water level (mAHD) | |
| --- | --- | --- | --- |
| | | 1978-1995 | 1996-2012 |
| 1. protect vegetation and mammals | SW < 21.6 | 21.66 | 21.39 |
| 2. prevent mosquitoes | SW < 21.6 | 21.33 | 21.41 |
| 3. protect waterbirds | SW < 21.6 | 21.84 | 21.44 |
| 4. protect frogs | SW < 21.6 | 22.02 | 21.61 |
| 5. protect tortoises | SW < 21.6 | 21.66 | 21.39 |
| 6. protect macro-invertebrates | SW < 22.0 | 21.66 | 21.39 |
| 7. prevent exposure of Acid Sulphate Soils | GW < 21.1 | 21.66 | 21.39 |
| 8. maintain sediment processes | GW < 21.1 | 21.66 | 21.39 |
















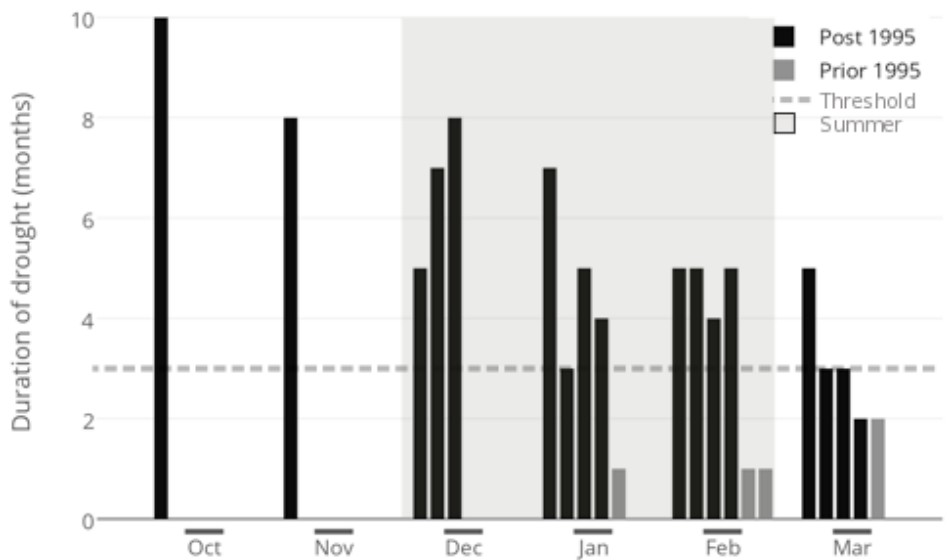








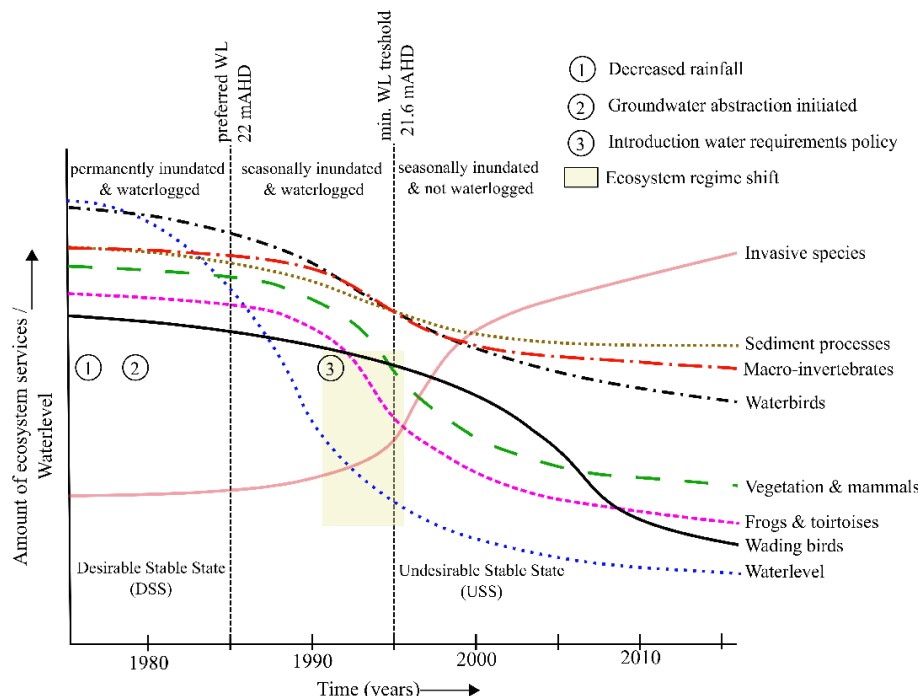
