# Peer review of "Adaptation tipping points of urban wetlands under a drying climate"

_Hydrology and Earth System Sciences, 2017_

## Referee Comment (RC1) · Anonymous Referee #1 · 20 Jun 2017

Overview

This paper uses adaptation tipping points (ATPs) as a method for measuring the effectiveness of wetland management strategies which can be used to monitor indicators as the climate changes. The research is undertaken using largely qualitative methods through desktop literature review, document analysis and interviews supplemented by measured indicators of exceedance of dryness indicators and objectives in management plans. The research objective is to develop an approach that can be used in data sparse situations and potentially under conditions of future climate uncertainties. The idea of the paper is a good one which builds on Kwadijk et al 2010 which shifts the sequence of inquiry from a top down approach to one where conditions of failure of policy objectives drives the monitoring system. The following revisions are necessary

for publication. 1) Tighten the argument with particular attention to the English writing style which is quite unclear in places and variable throughout. 2) The problem statement in the introduction is poorly written. The abstract is better and could be used in the text in the same sequence. The introduction is a sequence of literature rather than a discussion and leaves the reader unclear why certain issues are raise. For example, the terms 'thresholds' and 'tipping points' and then 'turning points' are used without any definitions as to how they differ. At line 94 the authors settle on ATPs but don't actually discuss the distinction from the other terms. One is left with the impression that all the terms are the same, but used in different contexts. My knowledge of the same literature is that these terms are in fact distinctly different, as discussed in the Werners et al 2013 paper. 3) Between lines 112 and 113 there is a jump in logic. What goes before does not in itself support ATPs as a suitable methodology for measure management strategy effectiveness. The lines 109 through 114 need to be recast to make sense. Later on at line 156 the statement is made that 'these observed impacts of climate change on the hydrology make the wetland a suitable study area to apply the ATP method.' Yet nowhere is this justified. It seems to be assumed that because there are climatic influences ipso facto ATP is a suitable methodology. The paper needs to be more specific about why this is so. What is it about ATPs that make it so? 4) At line 118 /119 the sentence suggests that there are three ecosystem functions and three aims. This is confusing. The sentence could say something like "effectiveness is defined using three ecosystem functions –hydrological response and variation/ temporal scale ecosystem responses/ recovery rate or alternative stable state of ecological processes. These were defined using literature review and interviews; hydrological time series data for each socio ecological objective from the management strategy; and minimum and maximum water level thresholds compared with mandated management objectives and policies respectively" 5) Similarly lines 130- 39 are quite confused. 6) Line 147 using Mediterranean climate as a descriptor for Western Australia seems to be written for a European audience. Could be deleted. 7) Line 154 refers to 'despite high resilience. . .' which is not supported by any evidence. Delete. 8) At line 202 the

Interactive
comment

term 'acceptable thresholds' is introduced with no mention of to whom and why they are or have to be acceptable. What follows at line 209 should be in the section 2 items. 9) Why the two time periods are created at 1995 is not mentioned? 10) The discussion section 331 -332 seems to restate the objective of the research in a new way. First you have set out that you are testing ATP as a method for monitoring the effectiveness of the management strategy. Now it is redefined to assess whether a baseline ecosystem management strategy was sufficient to sustain the ecological resilience of the ecosystem. This is a fundamental problem with paper because by the time the ATP is reached it is in all likelihood too late to act because there is a lead required to undertake the action and have it operate to maintain the ecosystem. So this means the ATP as a measure of management strategy effectiveness cannot operate except as and ex post description of what has happened, rather than a measure for prospective management. Fundamentally the authors need to address whether a different strategy could have halted the decline by running a stress test of different strategies for their effectiveness in maintaining the plans objectives. This is what the dynamic adaptive policy pathways approach (Haasnoot et al 2013) does. 11) At 372-374 there is a claim of the criticality of scale and level of policy and legislation and its relationship with shifting the social system, but there is no discussion of why this should be so. This could be elaborated and with reference to literature on scale (Oran Young could be useful here). The ideas are picked up again at line 391. Given the initial claim of linking the ATPs to the strategy this section is somewhat under done. What were the connections between the ecosystem function and the strategy- use examples of how the plans did this and whether the plans were monitored. 12) The sentence at 412-414 does not say how this could be done, i.e. 'by capturing the combined measures to adapt the current strategy.' What is meant by this? 13) Line 419 raises 'time periods' for when the policies expire. If this is prospective then putting time periods on would not be possible due to uncertainties about the rates and scale of the changes. All you can do is describe under what conditions the policies would fail. While you go on to raise the use of scenarios it is unclear whether the two sentences are connected. 14) Line

427-430 makes unsupported claims. These need to be discussed and supported. 15) The table with the timeline should be redrawn to extend the period when the thresholds were exceeded. Also there is a mix up of dates in the two lower right boxes as they relate to the timeline. In summary, the study is an interesting idea that potentially could build the body of knowledge but the paper needs significant rewriting to be clear and to fully discuss the link with the management strategy. It is also a post fact research inquiry, which begs the question as to how it can be applied prospectively. This could be further explored and clearly reported. The strength of the paper as conceived is its link with monitoring a management strategy, but no data is presented for the reader to see what the strategy sets out to do and whether the indicators were present in it and how the ATPs could be applied for uncertain future conditions before the ecosystem declines beyond the point of no return.

---

## Short Comment (SC1) · 12 Jul 2017

The paper explores a new method to assess effectiveness of management strategies, given a high degree of uncertainty. The presented case study deals with a wetland system in Australia. This is an important and much needed approach, given the frequent lack of data, which should be no excuse to abstain from the assessment of management effectiveness.

---

## Referee Comment (RC2) · Anonymous Referee #2 · 17 Jul 2017

This is an interesting study that uses the ATP framework to assess wetland management strategies under uncertainty. While there are some aspects of the paper that are promising, the paper requires a large amount of revision to be publishable. I have made some suggestions for improving the paper in the comments below.

- The paper centres on the authors' modified ATP framework, however the only deviation from the framework by Kwadijk et al (2010) seems to be that the authors did not complete all the original steps. To be able to present itself as a new modified method, the authors needs to provide a stronger argument for reducing the original approach and a better demonstration of its application (more suggestions below).

- The paper is quite difficult to follow due to its organisation and language. The paper needs to be carefully edited for grammar and choice of terms. Some parts of the paper

are repetitive or are not very informative, whilst other parts are unclear and missing key information (examples in minor comments below). This editing should have been done before submission, especially given the number of authors.

- Here are a couple of examples of inadequate descriptions for critical values/points of time, which were simply stated without any supporting evidence. For example, where did the minimum threshold values (21.6mAHD and 22.0mAHD in lines 226-229) come from? What values do they represent? How was 1995 identified as the critical point of change?

- In terms of the paper's structure, some of the text needs to be reorganised. Currently much of the text in Methods (particularly 2.2) belongs either before Methods (perhaps a new section describing the case study) or in the Results section.

- Given the aims of the paper, I expected the paper to have better demonstrated the ATP framework, by including a more detailed discussion (in Sections 4 and 5) about the effectiveness of the policy and management strategies in the case study, rather than simply stating that the policies were assessed.

- Table 2: It is unclear what the water levels (1978-1995 and 1996-2012) represent and how the ATPs were determined from Table 1 and Eq 1. More information is needed in the caption and text.

- Fig 5: this figure is quite confusing and a bit misleading. Firstly because of two different definitions of drought are given for the two contrasting periods (ie we are not comparing like with like). Secondly the layout is not logical (why are the "prior" years plotted after the "post" years") - what do the secondary x-axis represent? The summer months are highlighted, presumably to show that the drying of the wetland in critical periods - however, if my interpretation of the plot is correct, wouldn't the two bars in Oct (10 months) and Nov (8 months) also represent periods when the drought extended over the critical summer period? It seems the main point of Fig 5 is to show that drought frequency has increased since 1995. It would make more sense to me to have

a figure that simply shows inundation as a time series instead - this would more clearly show that the wetland has shifted from a being permanently to seasonally inundated than the current plot.

- Section 4.1/line 332-333: the authors claim that their framework assesses resilience of the hydrological system across spatial and temporal scales. How were spatial scales addressed in the framework?

- One of the critical issues with the approach/framework relates to the identification of the ATP in time. Often it is not known whether a tipping point/threshold has been crossed until after the fact (this is somewhat implied by the authors e.g. lines 65-72). Furthermore, there are time lags associated with management and ecological responses (this was only mentioned by the authors in the very last sentence of the paper). These issues and their implications on the approach/framework need to be discussed.

Minor comments: - Line 154 "Despite high resilience, the wetland shows a rapid decline.." This sentence is contradictory.

- lines 296-298: The values 5x and 16x and their definitions do not make sense.

- lines 300-303: although contrary to regulation, it is completely logical that drying is more likely to occur over summer.

- line 321-322: what is meant by "existing water requirements"?

- line 370-372: this sentence contradicts itself. Firstly it implies that the management interventions were not triggered due to a gradual transition in the system, but on the other hand it described the processes in the ecosystem as being rapid.

- line 372-374: What is the evidence of this claim? The author has provided no information that suggests that shifts in the social system stem from the scale and level of policy and legislation.

- The discussion seems to frequently shift between talking about the specific case study to talking about cases in general, without clarification. Eg. line 377-381: Are these ineffective policies for the same study area or elsewhere?

- line 401-402: This sentence is not informative - it simply says reversing conditions in the ecosystem requires measures that reverse the conditions.

- line 421: What do you mean by "coupled system"?

- line 418-420: This sentence states that "With the involvement of stakeholders in our assessment we can account for the exploration of future hydrological events and provide decision-makers time periods for when the expiry of current policies occur?" This did not seem to be done in this current study. Furthermore, given that the authors modified framework only examines historical data, it seems the original ATP assessment frameworks is needed to achieve those outcomes, not the authors' version.
* * *

---

## Short Comment (SC2) · 9 Aug 2017

Thank you for your interest in this research project. We aim in the second phase of the research to address the challenges that stakeholders face when adaptive ecosystem management is required.

---

## Author Comment (AC1) · 9 Aug 2017

Thank you for your interest in the research and suggestions for improving the paper. We accept the minor comments and I respond to the major comments below.

Comment 1: Review English language

Comment 2: Due to the focus on ecosystems and the different literature covered in the introduction for adaptation; we decided to refer to existing literature for the different definitions of thresholds, tipping points, and turning points (Werners et al 2013).

We propose to address the above comments as follows in lines 73-82: Thresholds and tipping points are important focal points for adaptive management (Folke et al. 2005;ÂăRijke et al. 2012; Haasnoot et al. 2013; Werners et al. 2013), but often lack

data to define exact biophysical thresholds to model complicated interactions in SES models (Schlueter et al. 2012). However, several indicators (Niemi and McDonald 2004) and policy-based approaches do exist that are commonly used in flood mitigation (Lavery and Donovan 2005; Kwadijk et al. 2010; Reeder and Ranger 2011; Gersonius et al. 2012), water resources management (Brown et al. 2011; Poff et al. 2015), and institutional adaptation (Lawrence et al. 2013, Fünfgeld 2015) to determine when the boundaries of a system are reached and future change becomes critical for the system. Despite the extend of literature, there is limited focus on defining thresholds for ecosystem processes and informing policies to how change has become critical (Hanger et al. 2013) and when interventions are needed to address different key ecosystem processes.

We propose to address the above comments as follows in lines 83-85: A policy-based approach that defines when and which objectives of a current strategy are not reached, is a starting point to adapt existing and formulate new strategies (Kwadijk et al 2010). An 'adaptation tipping point' is the moment when the magnitude of change is such, that a current management strategy can no longer meet its objectives.

Comment 3: Proposed amendment to line 112-114: There is a high expectancy that policies are inadequate and management authorities have the desire to understand past effects of climate change on maintaining individual socio-ecological objectives. With already a management plan available, multi-scale policies in place, and limited data availability, the wetland is suitable to apply the ATP methodology.

Comment 4: We will define effectiveness more clearly by accepting the helpful comments.

Comment 5: This paper aimed to address the difficulties to determine ATPs for ecosystems. These include to overcome a lack of data to inform management, but also addresses how to deal with management objectives (across scales) and the determination of threshold values (only partly available) that represent ecosystem processes.

[Figure]

Therefore, to limit our focus we have adapted the original ATP methodology to only determine ATPs. We propose the following amendments to this section: Keep lines 131-134 with reference to the original methodology. Lines 134-135: We modified the original methodology to determine ATPs for different socio-ecological objectives and thresholds with the assessment of historical hydrological time series. We expanded step 3 to interpreted ATPs in conjunction with the hydrological response and variation; temporal scale ecosystem responses; and recovery rate and alternative stable state of ecological processes (Figure 1).

Comment 6: We will keep 'Mediterranean climate' as a descriptor, since Western Australia compromises of distinct climatic zones not every reader would be familiar with.

Comment 7: Delete sentence

Comment 8: 'Acceptable policy thresholds' refer to mandated Ministerial water requirements cited in line 217. Change to 'Mandated policy thresholds'. Line 209 moves to section 2 items.

Comment 9: With Eq.1 we calculated the frequency of exceedance of water levels stated in Table 1 from observed time series that were divided in two time periods (cited and explained in line 232). Each period reflects the time period for policy measures generally to be adapted.

Comment 10: We agree that 'baseline' is confusing in the context of this paper. This paper aimed to address the difficulties to determine ATPs for ecosystem and the baseline refers to existing policies and management objectives. Due to the extend and ongoing research we decided to focus on only determining ATPs. We are currently preparing research to determine alternative strategies with stakeholders to postpone or eliminate existing ATPs according to the steps of the original ATP methodology. We will also use the dynamic adaptive policy pathways approach (Haasnoot et al 2013). This will be reported in a separate manuscript.

Comment 11: There may be no shift in the social system, however, we provide a discussion among management authorities to consider management objectives and threshold values. The management objectives are derived from different sources such as the State-scale water level criteria; the national (Commonwealth) ecological objectives that are linked to the Ramsar guidelines; and the key socio-ecological objectives from the local management plan. Currently, National and state policies are monitored with only partly the policies determined on local scale. We propose to establish a clearer link (lines 372-376) to the scale mismatch of policies and the ATPs that were determined. Also, we propose to discuss ATPs and the possibilities of interventions on different scales when stakeholders act for different socio-ecological objectives(lines 387-391).

Comment 12: In the absence of a combined eco-hydrological and social model we could include ecosystem-based and policy-based adaptation measures to adapt strategies.

Comment 13: We describe under what different conditions policies fail. These include to overcome a lack of data to inform management, but also addresses how to deal with management objectives (across scales) and the determination of threshold values (only partly available) that represent ecosystem processes.

Comment 14: We propose to reword the sentences. (Lines 427-430) Despite the exceedance of critical thresholds, management has not responded adequately to changing hydrological variation of the ecosystem. We assumed climate change to be the main external driver for the ecosystem regime shift. The ATP application is adequate for ecosystems when a clear external driver of change can be determined; stakeholders agree on setting thresholds; and expand individual management objectives to objectives across several levels of policies. However, the study becomes complicated when multiple stressors are responsible for ecosystem change and stakeholders do not include objectives or thresholds defined by different policies.

Comment 15: We will expand the orange box and resize the blue box accordingly to rectify the confused dates in the lower orange and blue boxes.

---

## Author Comment (AC2) · 9 Aug 2017

Thank you for your interest in the research and suggestions for improving the paper. We accept the minor comments and I respond to your major comments below.

Comment 1: This paper aimed to address the difficulties to determine ATPs for ecosystems. These include to overcome a lack of data to inform management, but also addresses how to deal with management objectives (across scales) and the determination of threshold values (only partly available) that represent ecosystem processes. Therefore, to limit our focus we have adapted the original ATP methodology to only determine ATPs. In our last comment, we refer to preparation of research to determine alternative strategies with stakeholders to postpone or eliminate existing ATPs

[Figure]

according to the steps of the original methodology.

We will change sections in the introduction: Add to lines 73-82: Despite the extend of literature, there is limited focus on defining thresholds for ecosystem processes and informing policies to how change has become critical (Hanger et al. 2013) and when interventions are needed to address different key ecosystem processes.

Add to lines 83-85: A policy-based approach that defines when and which objectives of a current strategy are not reached, is a starting point to adapt existing and formulate new strategies (Kwadijk et al 2010). An 'adaptation tipping point' is the moment when the magnitude of change is such, that a current management strategy can no longer meet its objectives

We propose the following amendments in the method section: Keep lines 131-134 with reference to the original methodology. Lines 134-135: We modified the original methodology to determine ATPs for different socio-ecological objectives and thresholds with the assessment of historical hydrological time series. We expanded step 3 to interpreted ATPs in conjunction with the hydrological response and variation; temporal scale ecosystem responses; and recovery rate and alternative stable state of ecological processes (Figure 1)

Comment 2: Threshold values are stated in the Ministerial water requirements policy cited in line 217. The values represent the height of the water table expressed in mean Annual Height Datum (mAHD) with 21.6 mAHD representing the lake bed. A value of 22.0 mAHD represents a water depth of 40cm. No exact critical point of change can be determined, however, during the 1990s the lake's hydrology altered (more frequent dry spells, longer drought duration). This is consistent with observations in other studies cited in the discussion section.

Comment 3: We propose to move section 2.2.1 (lines 162-177) to section 2.1 and remove the redundant text in lines 140-157. Section 2.1 will be renamed to 'case study area'. Lines 178-187 move to section 2.1 and redundant text in lines 147-157 will be

removed. Delete lines 188-193 and mention in discussion section. Move lines 193-194 and figure 4 to results section. Keep section 2.2.2 and add refer to case study area (section 2.1) as step 1 of methodology. Keep section 2.2.3 in methods section.

Comment 4: We consider including a detailed discussion about the scale differences of current policy and management strategies in section 4.1. In section 4.2 we will add how different management authorities are responsible for different policies and how the ATP methodology helped to bridge the different views of stakeholders that are involved in the execution of the management strategy. We propose to include in section 5 the importance of reviewing a range of policies to enable discussion among stakeholders to determine existing and new management objectives/thresholds.

Comment 5: We will extend the caption and text with a reference to threshold values that are stated in the Ministerial water requirements policy cited in line 217; and that values represent the height of the water table expressed in mean Annual Height Datum (mAHD) with 21.6 mAHD representing the lake bed. A value of 22.0 mAHD represents a water depth of 40cm. With Eq.1 we calculated the frequency of exceedance of water levels stated in Table 1 from observed time series that were divided in two time periods. (cited and explained in line 232). Each period reflects the time period for policy measures generally to be adapted.

Comment 6: We will change the figure. Prior and post will be reversed and the threshold definitions will be mentioned in the figure. In this way, we can still show that the lake has dried prior to 1995, but not considered to be a dry period according to the policy definition.

Comment 7: Prior and post will be reversed. Also add 'time' to x-axis.

Comment 8: That is correct and this indeed causes confusion. Policy requires drying not to occur before April/May. We will remove the summer period and instead mark April/May in the figure as the earliest moment for the lake to dry that is currently allowed.

Comment 9: Yes, that's correct. We also tried to explain how the drought frequency increase is related to the policy objectives. Therefore we included the start month of a dry period and the duration of each dry period. We also tried to include that drying of the lake occurred prior to 1995, however, too short according to the policy definition of 3 consecutive months.

Comment 10: The shift from permanently to seasonally inundated has already been shown in previous studies, including such a time series plot. We address the shift also in Figure 7.

Comment 11: We included ecological objectives from the local management plan and Ramsar objectives. Water level thresholds are determined by State level government and defined according to regional scale hydrology of the wetland systems of the Swan Coastal Plain. We aim to make a clearer connection of policies across scales in the revised methodology section (Section 2.2.2) and stakeholder representation that followed from the institutional organisation.

Comment 12: The timing of an ATP cannot be determined precisely and for ecosystems need to be interpreted with other data, such as bird counts, macro-invertebrates counts/species composition, and number of weeds. Suggestion for change: Lines 363-371: We will add that management responses take time. When ATPs have occurred, these do not immediately translate in ecological degradation due to lag response of ecological processes (as a result of water decline). The current framework does not account for ecological lag response and the time that is needed to implement new actions. Lines 353-354: We combine the available ecological data with ATPs which could be used to prioritise management actions.

Comments 13 and 14: Delete sentence

Comment 15: accept

Comment 16: Change to: "water level requirements under groundwater management

policy" (cited in line 217)

Comment 17: Change sentence to: "Management interventions were not triggered due to gradual water level decline over several decades. However, the observed ecosystem shift occurred in a relative short period."

Comment 18: No shift in the social system, however, provides a discussion among management authorities to consider management objectives and threshold values.

Comment 19: Ineffective policies in case study area (line 376-377) and compared to other studies (lines 377-381)

Comment 20: Delete sentence

Comment 21: Without a combined eco-hydrological and social model that captures the feedbacks between these two domains.

Comment 22: Change lines 428-420: future research could assess... We are currently preparing research to determine alternative strategies with stakeholders to postpone or eliminate existing ATPs according to the steps of the original methodology. This will be prepared as a separate manuscript. This paper aimed to address the difficulties to determine ATPs for ecosystems. These include to overcome a lack of data to inform management, but also addresses how to deal with management objectives (across scales) and the determination of threshold values (only partly available) that represent ecosystem processes.